# Brassinosteroids Regulate the Water Deficit and Latex Yield of Rubber Trees

**DOI:** 10.3390/ijms241612857

**Published:** 2023-08-16

**Authors:** Bingbing Guo, Mingyang Liu, Hong Yang, Longjun Dai, Lifeng Wang

**Affiliations:** Key Laboratory of Biology and Genetic Resources of Rubber Tree, Ministry of Agriculture and Rural Affairs, Hainan Key Laboratory for Cultivation & Physiology of Tropical Crops, State Key Laboratory Incubation Base for Cultivation and Physiology of Tropical Crops, Rubber Research Institute, Chinese Academy of Tropical Agricultural Sciences, Haikou 571101, China; guobingbing1989@126.com (B.G.); liumy2023@126.com (M.L.); yang_hong0317@126.com (H.Y.); ldailongjun@126.com (L.D.)

**Keywords:** Brassinolide, water deficit, latex, natural rubber, rubber tree

## Abstract

Brassinolide (BR) is an important plant hormone that regulates the growth and development of plants and the formation of yield. The yield and quality of latex from *Hevea brasiliensis* are regulated by phytohormones. The understanding of gene network regulation mechanism of latex formation in rubber trees is still very limited. In this research, the rubber tree variety CATAS73397 was selected to analyze the relationship between BR, water deficit resistance, and latex yield. The results showed that BR improves the vitality of rubber trees under water deficit by increasing the rate of photosynthesis, reducing the seepage of osmotic regulatory substances, increasing the synthesis of energy substances, and improving the antioxidant system. Furthermore, BR increased the yield and quality of latex by reducing the plugging index and elevating the lutoid bursting index without decreasing mercaptan, sucrose, and inorganic phosphorus. This was confirmed by an increased expression of genes related to latex flow. RNA-seq analysis further indicated that DEG encoded proteins were enriched in the MAPK signaling pathway, plant hormone signal transduction and sucrose metabolism. Phytohormone content displayed significant differences, in that trans-Zeatin, ethylene, salicylic acid, kinetin, and cytokinin were induced by BR, whereas auxin, abscisic acid, and gibberellin were not. In summary, the current research lays a foundation for comprehending the molecular mechanism of latex formation in rubber trees and explores the potential candidate genes involved in natural rubber biosynthesis to provide useful information for further research in relevant areas.

## 1. Introduction

Brassinosteroids (BRs) belong to the class of sterol compounds and were first discovered in pollen from rapeseed. They are now recognized as the sixth plant hormone [1,2]. Although the content of BR is very low in plants, the physiological activity is very high. A great deal of research indicates that almost all of a plant’s organs can synthesize BR to regulate seed germination, flowering, senescence, topic growth photosynthesis, stress resistance, etc. BR is also closely related to other signaling molecules [3,4,5,6,7,8]. The application of 2,4-epbrassinolide (EBL) was shown to alleviate Ca (NO_3_)_2_ stress in cucumbers by regulating the absorption and distribution of mineral nutrients [9]. Additionally, EBR can increase the inhibition of photosynthesis, the activities of antioxidase and Rubisco activase (RCA), and the gene expression in wheat induced by drought and heat stress [10]. BR has been shown to alleviate oxidative damage and improve hypocotyl length, biomass, and sprout quality in radishes stored at low temperature by increasing the activities of POD and SOD [11]. EBR improved drought tolerance of tobacco by increasing the activities of SOD, POD, and CAT enzymes and the expressions of their related genes, along with a higher accumulation of osmoregulatory substances [12]. Additionally, BR promotes photosynthesis and growth of plants by regulating the synthesis and activities of various photosynthetic enzymes in cucumber leaves and can also increase photosynthesis by regulating the expression of Rubisco subunits and other Calvin cycle enzyme genes [13]. Plant photosynthesis provides the substrate and energy supply for the synthesis of plant primary metabolites. The effect of BR in plants differs with concentration, and the appropriate BR concentration can increase chlorophyll contents and photosynthetic efficiencies in cucumbers [14]. The BR signaling pathway is one of the best understood signal transduction pathways, and its interactions with plant hormones are well known [15,16]. BR is involved in auxin synthesis, transport and signal transduction, and promotes the polar transport of auxin to regulate its distribution in different plant organs and tissues [17]. Exogenous auxin treatment can also up-regulate the expression of *DWF4*, which is a BR synthesis gene, and thus increase the concentration of endogenous BR [18]. Moreover, BR regulates the concentration of endogenous GA in plants. GA promotes cell elongation and requires the response of BR along with BZR1, a transcription factor of BR combined with the promoter of the GA metabolism gene *GA20x* [19,20]. BR also positively regulates ethylene synthesis during cell elongation, and pollen tube-sensing is related to the up-regulation of the key enzyme of ethylene synthesis (ACC synthase) by BR [21,22].

Natural rubber (NR) is produced from rubber trees. With its good mechanical and insulating properties, among others, is one of the most indispensable important raw materials in industrial production, daily life, and agricultural, medical, and other industries; almost all industrial products are inseparable from the production of natural rubber. BR is widely used in production, and its results and reactions are different under different conditions and between different species. A lower concentration of BR solution can trigger a stronger physiological response in plants, but it has not yet been used in rubber trees. It is known that the receptor and transcription factors of the BR signaling pathway are expressed in latex and induced by phytohormones [23,24], so we suspected that BR also has some physiological functions in rubber trees. Latex originates from sucrose produced by photosynthesis on the leaves of rubber trees. Latex flow requires a lot of water and leaves the rubber tree with a water deficit. In this research, we applied BR to rubber trees for the first time and found that BR can increase photosynthetic index, antioxidase activity, and osmotic regulation to alleviate the water deficit of rubber trees. We also analyzed the transcriptome, aiming to explore the significance of BR in yield formation in rubber trees, and found that BR can also improve latex yield and the molecular weight of rubber hydrocarbons by increasing the expression of NR biosynthesis genes and decreasing the expression of latex-plugging genes. This result will provide a reference for the research and development of novel yield regulators and can help to guide the sustainable development of the rubber industry.

## 2. Results

### 2.1. Effects of Exogenous BR on Photosynthetic Characteristics of Rubber Tree Leaves under Water Deficit

Water deficit (WD) is an important stress factor limiting latex yield in rubber trees. Under conditions of WD, rubber trees produced more wilted, yellow, and fallen leaves versus the control. With BR treatment, the effect of water deficit on rubber trees was weakened (Figure 1A). The rates of photosynthesis in rubber trees in the control, WD, and WD with BR groups were determined by measuring Chl fluorescence (Figure 1B and Appendix A). The maximum quantum yield of PSII (Fv/Fm) and maximum capture efficiency of PSII (Fv’/Fm’) were decreased under WD. After BR treatment, these two parameters became similar to those in the control plants, even under WD conditions. Photochemical quenching (qP) and nonphotochemical quenching (NPQ) are important components of photosynthetic quantum efficiency regulation in plants. They displayed opposite trends in rubber trees under WD. qP decreased and NPQ increased as processing time increased under WD, but BR reduced these changes. Electron transfer efficiency (ETR) reflects the apparent electron transport efficiency under the actual light intensity. It decreased under WD. WD destroyed the photosynthetic mechanism of rubber trees, decreased the ETR and qP, reduced the utilization rate of light energy, and led to a decrease in photosynthesis. However, BR reduced the decrease in light use rate and efficiency in rubber trees under WD conditions.

### 2.2. Effects of Exogenous BR on Physiological Characteristics of Rubber Tree Leaves under Water Deficit

With the extension of WD time; leaf relative conductivity (REC); and content of osmotic adjustment substances such as malondialdehyde (MDA), proline (Pro) and hydrogen peroxide were significantly increased compared to the control plants (Figure 2A). Exogenous BR reduced the permeability of the cell membrane under WD conditions, decreased the contents of MDA and REC, further promoted the accumulation of proline in the cytoplasm, and decreased the content of hydrogen peroxide in the cytoplasm. Water deficit was accompanied by a decrease in the content of energy substances and sucrose, and the rate of energy biosynthesis and sucrose content in cells was greatly increased with the addition of BR (Figure 2B). As time progressed under WD treatment, antioxidase activity presented different trends. APX, CAT, and SOD activities increased for the first 6 days before falling, whereas POD activity increased constantly. BR enhanced the activity of antioxidases, further reducing oxidative damage (Figure 2C). As shown in Figure 2D, WD induced the transcription of antioxidase genes with the same pattern: a decrease for the first 3 days, then an increase until day 6, and, finally, a decrease at day 9. Under exogenous BR under WD conditions, all antioxidase-related genes were up-regulated to repair the cell damage caused by ROS. Primers used in this section are shown in Appendix A.

### 2.3. Effect of BR on Latex Physiological Parameters of Rubber Trees

The biochemical and physiological parameters of latex are closely related to latex regeneration and latex flow and are used to evaluate the strength of rubber exploitation. When exogenous BR was applied, the latex yield, yield increase rate, dry rubber yield, lutoid bursting index, and total solid all increased, with the yield increase rate increasing to 133%. Plugging index and initial velocity were decreased by BR, which extended the length of latex flow time and increased yield. Dry rubber content, mercaptan, inorganic phosphorus, pH, and sucrose did not significantly vary when BR was applied to the rubber trees. Furthermore, BR increased average molecular weight and reduced the dispersion coefficient, improving latex quality (Figure 3A,B). We simultaneously investigated the regulatory effects of BR on expression levels of genes related to rubber biosynthesis, rubber hydrocarbon chain elongation, and latex-plugging-related genes and discovered that the expressions of genes *HbFPS*, *HbHMGR*, *HbGGPPS*, *HbSRPP*, *HbHRT*, and *HbREF*, which participated in rubber biosynthesis initiation and rubber hydrocarbon chain elongation, were increased, which facilitated the formation of higher quality latex. *HbHEV*, *HbHGN*, and *HbCHI* are associated with latex plugging to prevent latex flow; BR application inhibited the expression of these three genes and increased latex yield by prolonging the extraction time of latex (Figure 3C). Primers used in this section are shown in Appendix A.

### 2.4. GO Enrichment and KEEG Pathway Enrichment Analysis of DEGs

In order to analyze the transcriptomes of rubber latex induced by BR, RNA was isolated from rubber latex and treated without BR as a control. Differentially expressed genes (DEGs) were confirmed after BR treatment (FDR < 0.05, *p* < 0.05). DEG analysis indicated that 1213 DEGs in the latex of rubber trees showed altered expressions in response to BR, with 751 up-regulated genes and 462 down-regulated genes (Figure 4).

GO enrichment analysis found that DEGs in latex after BR treatment were concentrated in the three categories of biological processes (BPs), cell components (CCs), and molecular functions (MFs) (Figure 5). The majority of DEGs in the BP category were found in the metabolic process, cellular process, and signal–organism process subcategories. In the CC category, most of the DEGs belonged to the cell, cell parts, and membrane subcategories. In the MF category, DEGs were mostly involved in binding, catalytic activities, nucleic acid binding transcription factor activities, and other activities.

KEGG enrichment was used to investigate the possible biological functions of DEGs identified with BR in latex. The top 20 KEGG pathways are shown in Figure 6A, and statistics of KEGG pathway enrichment are shown in Figure 6B. The results indicate that most of the 119 pathways involved in this study participate in the MAPK signaling pathway and in protein processing in the endoplasmic reticulum.

### 2.5. Difference Analysis of Metabolite Accumulation

The KEGG pathway analysis demonstrated that metabolites in this study were involved in fifty pathways, the majority of which involved metabolism. Genes were involved in sugar and sucrose metabolism, amino acid biosynthesis, phenylpropanoid metabolism, flavonoid metabolism, and pyruvate metabolism (Figure 7A).

The results of various phytohormones in latex treated with or without BR are shown in Figure 7B. Based on these results, we can see that the levels of phytohormones related to ethylene synthesis, salicylic acid synthesis, and some cytokinins were up-regulated, whereas other hormones, such as auxin, gibberellin, cytokinin, abscisic acid, and alkaloid, were down-regulated by the BR stimulus.

## 3. Discussion

The rubber tree is a member of the Euphorbiaceae family and is an important economic plant grown in tropical Asia. As a strategic resource, the only source of natural rubber is the latex that flows from rubber trees [25]. Phytohormones are chemical messengers that occur in very low concentrations in plants [26]. The distribution of phytohormones in different tissues determines the characteristics of plant growth and development. At the same time, the allocation and transport of phytohormones under specific physiological conditions can improve the adaptability of plants to the environment [27]. The application of several phytohormones to rubber trees has been frequently reported, such as ethylene, abscisic acid and jasmonic acid, which are related to the growth and resistance of rubber trees [28,29,30]. In this study, we determined the role of BR in regulating water deficit and latex yield in rubber trees.

The basic foundation of growth for plants is photosynthesis. The net photosynthetic rate of plants will decrease, and stomata will close when plants are subjected to water deficits, preventing normal growth and development [31]. Photosynthetic parameters reflect a lot of information about the electron transport process of PSⅡ photosynthesis. Previous studies have indicated that the reason why the primary light energy conversion efficiency Fv/Fm of PSⅡ decreased after plants were subjected to abiotic stress was that the light energy absorbed by the photosynthetic organs exceeded the amount of energy they could use, resulting in light inhibition, which led to the damage of PSⅡ [32,33]. The results of this experiment showed that exogenous BR alleviated the decrease in the primary light energy conversion efficiency (Fv/Fm) in rubber tree leaf buds caused by a water deficit, bringing it close to the levels in the control. This indicates that BR can significantly alleviate the damage caused to PSⅡ in rubber tree leaf buds under a water deficit and is consistent with the research results of other scholars [34,35].

As a semi-permeable membrane, the cell membrane plays an important role in the intra/extracellular exchange and utilization of substances, and the REC can assess the degree of plant damage [36]. Water deficit destroys the stability of the cell membrane, resulting in the exosmosis of the electrolytes in the cell and an increase in the REC. With the spraying of BR, REC decreased significantly, indicating that exogenous BR alleviated the changes in membrane permeability caused by water deficit, thus reducing the REC. The content of Pro is often regarded as an indicator of plant stress resistance. Our results showed that water deficit caused protein decomposition with the extension of stress time, releasing proline and thus sharply increasing the Pro content. After application of BR, the Pro contents in leaves were significantly higher than that in the control, which may have been attributed to the direct involvement of exogenous BR in plant physiological growth by promoting Pro synthesis, which was in keeping with previous studies [37,38].

Under stress, the dynamic balance of ROS production and clearance in plants is broken, affecting the content of MDA, and biofilm lipid peroxidation is triggered and intensified, thus damaging the plant [39]. Based on the findings of a study on mini Chinese cabbage, in which BR-mediated protein S-nitrosylation alleviated low-temperature stress, the accumulation of MDA content decreased with BR treatment, eliminating ROS as an antioxidant substance, inhibiting membrane lipid peroxidation and enhancing the stability of the membrane, so as to alleviate the oxidative damage caused by water deficit on rubber tree buds [40]. SOD, POD, CAT, and APX play an important role in preventing ROS toxicity [41]. In our study, the activities of SOD, POD, and CAT first increased and then decreased with the extension of stress treatment, due to the fact that plants spontaneously regulate physiological activities in the early stage to enhance SOD, POD, APX, and CAT activities and remove ROS. However, if the stress continues or intensifies, plants cannot perform spontaneous regulation, resulting in decreased enzyme activity and increased oxidative damage to the plant. The gene expressions of *HbAPX*, *HbCAT*, *HbPOD*, *HbMnSOD*, and *HbRbsS* support this suggestion. With the processing of BR, the activities of SOD, POD, APX, and CAT were significantly higher than those without BR treatment, showing that BR enhanced the defense ability of the plant antioxidant system and maintained the metabolic balance of the intracellular ROS system, thus alleviating the oxidative damage caused by water deficit. This result confirms previous studies that showed that BR-induced stress tolerance is associated with increased accumulation of ROS [42,43].

Rubber biosynthesis and latex flow are key factors determining the quality and yield of natural rubber [44]. Latex flow is accompanied by the loss of water, which can cause serious tapping panel dryness (TPD) [45]. Ethephon (an ethylene release agent) is used as a stimulant in tapping and can significantly increase latex yield by promoting the transport of water from the surrounding tissues into the laticifers, thus maintaining laticiferous turgor pressure, improving the stability of lutoids, and delaying the coagulation of latex [46,47]. In this study, we explored the potential function of BR in rubber biosynthesis and latex flow. BR increased latex yield by reducing plugging index and extending the course of latex flow without changing the dry rubber content, whereas ethephon increased latex yield by extending the duration of latex flow but significantly decreased dry rubber content. BR also increased the lutoid bursting index associated with latex coagulation, resulting in decreased initial velocity. However, there was no significant relationship with the main factor that determined the yield of latex flow time. Therefore, there was no direct relationship between BR stimulation and the integrity of the lutoid with another regulating mechanism regulation clearly involved. We suspect that BR improves latex yield by increasing the lutoid bursting index and reducing the plugging index by decreasing the expression levels of *HbHGN*, HbCHI, and *HbHEV*, which prevents latex flow progress. The molecular weight of natural rubber is heterogeneous or polydisperse. The average molecular weight is used as the characterization of natural rubber molecular weight, and the molecular weight distribution is used to characterize the degree of dispersion in natural rubber dispersion [48], which are important parameters used to measure latex quality. BR improved the Mn but caused no difference in Mw, reducing the dispersion coefficient (represented by Mw/Mn), thus improving the quality of natural rubber. *HbFPS*, *HbHMGR*, and *HbGGPPS* are related to NR biosynthesis, and *HbSRPP*, *HbHRT* and *HbREF* are related to rubber hydrocarbon chain extension and enhance molecular weight, which was increased by BR in latex. Physiological indicators and the expression of related genes reconfirmed our conclusions that BR improves the yield and quality of latex.

Plant metabolism, growth, development, and response to exogenous stimuli are all dependent on the perception and signaling of phytohormones [49]. In general, phytohormone response is the result of multiple pathways interacting with each other [50,51,52]. BR signaling may serve a function by interacting with other major phytohormones such as ethylene, gibberellin, cytokinin, salicylic acid, jasmonic acid, and abscisic acid [53,54,55,56,57,58]. The data obtained by RNA-Seq technology not only provide useful information for predicting the DEGs under BR treatment but also help to understand the transcriptional regulation of the plasticity of natural rubber production and latex flow. Based on GO and KEGG enrichment annotation, we measured the amount of DEGs enriched in phytohormone signal transduction, sucrose metabolism, and plant–pathogen interaction. We found that the transcriptions of CTR1, NAC, ABR1, ERF1, and ARG2—related to ethylene—TIFY and MYC2—related to jasmonic acid—PP2C, GEM, and ABI1—related to abscisic acid, IAA1/4/18—related to IAA—and SABP2—related to salicylic acid—were significantly increased by BR. Although it has been reported that all these phytohormones are related to growth regulation in a specific way [59], how BR coordinates the interactions with and stimulation of other phytohormones during the stimulation of latex quality and yield remains to be clarified. Transcription factors play an important role in many biological processes in a cell or organism by regulating gene expression. In addition, transcription factors play essential roles in the interactions between phytohormone signaling pathways [60,61]. In total, 1213 DEGs were identified in this study, with the largest transcript factor family being MYB, followed by MYB-like, bHLH, WRKY, and zinc finger proteins. Although it is not yet known whether their functions are related to BR stimulation of latex production, these data will be a meaningful resource for screening and identifying candidate genes involved in the regulation of complex signaling networks in BR response processes.

## 4. Materials and Methods

### 4.1. Plant Materials and Treatment

Buds from the rubber tree variety ‘CATAS73397′ were grown in soil in plastic pots at an experimental farm in the Chinese Academy of Tropical Agricultural Sciences in Danzhou City, Hainan Province, China (19°51′51 N; 109°55′63 E), with an average temperature of 30 °C, precipitation 180 mm, and humidity at 97.5% during the growing season to treat with water deficit. Under natural conditions, we set up three treatments (control, water deficit, and water deficit with BR) with three repeats over 9 days.

Bearing trees of ‘CATAS73397′ were also planted at the same farm with the s/2 d/4 system. We used 5 μmol/L BR, based on results from a previous study [23,24]. We divided rubber trees into two groups: control and BR treatment with three repeats. BR treatment was carried out according to previous results [62]. In the control group, we removed glue lines with tweezers, harvested flesh latex in ice box and frozen latex in liquid nitrogen, and then used a brush to evenly apply water to the cutting surface, before harvesting flesh latex again on the third day. In the treatment group, all steps were the same as in the control, except the solution used was 5 μmol/L BR instead of water. Latex samples were harvested at 0 h and 48 h, with flesh latex stored in a fridge and frozen latex in liquid nitrogen.

### 4.2. Determination the Physiological Parameters of Latex

The latex yield was measured as the total amount of latex that could be collected until latex flow was terminated. Total solid content was measured by drying five grams of latex at 100 °C to a constant weight and then calculating the ratio of dry weight to fresh weight. To determine the dry rubber content, 1 mL latex was taken from each sample and solidified with 5% glacial acetic acid, then rinsed with water overnight and dried to constant weight in an oven at 60 °C. Dry rubber yield was calculated by multiplying the dry rubber content by the latex yield. The initial velocity and plugging index were calculated by measuring the latex yield 0~5 min after tapping. pH was measured using a FiveEasy Plus FE28 pH/mV bench instrument (Mettler Toledo, Shanghai, China). Mercaptan, lutoid bursting index, inorganic phosphorus, and sucrose were determined immediately after tapping, according to the method followed in previous studies [44,63,64]. Molecular weight and dispersion coefficient were estimated using Gel Permeation Chromatography (GPC) 1515 (Waters, Milford, MA, USA), weighed and dissolved about 30 mg of latex sample in 10 mL tetrahydrofuran (THF), stood for 72 h, leached the supernatant with 0.45 μm filter, and then loaded the machine with the following parameters: internal detector temperature, 40 °C; external detector temperature, 40 °C; flow rate, 1.0 mL/min; and injection time, 25 min.

### 4.3. Observation of Photosynthetic Fluorescence Parameters in the Leaves of Rubber Trees

Based on the methods of Wang’s research [65], a PAM-2500 high-performance chlorophyll fluorometer (Walz, Effeltrich, Germany) with the data acquisition software PamWin-3 (Heinz, Walz, Düsseldorf, Germany) was used in this study to measure in vivo chlorophyll in darkness prior to and after 30 min. While the plant was adapting to the dark, minimal fluorescence level (Fo) was measured, which was low and without any significant variable fluorescence. Afterwards, a 0.8 s saturation pulse at 12,000 mmol m^−2^s^−1^ was given to determine the maximal fluorescence level of PSII centers once closed and after the leaves had adapted to the dark. Maximum fluorescence and initial minimum fluorescence were measured in dark-adapted leaves to calculate the maximum photochemical efficiency (Fv/Fm) of photosystem II. The numerical values of Fo, Fm (maximal fluorescence yield), Fo’ (initial fluorescence under light adaptation), Fm’ (maximum fluorescence under light adaptation), Fv/Fm (PS II), ETR (relative electron transport rate), Y(II) (actual quantum yield of PS II), Y(NPQ) (regulatory dissipation), and qL (photochemical quenching) were measured with a PAM-2500 high-performance chlorophyll fluorometer. Other parameters were calculated based on the measured values.

### 4.4. Observation of Physiological Parameters of Rubber Trees

Leaf samples were collected and used immediately. MDA, Pro, H_2_O_2_, ATP, sucrose content, and APX, CAT, POD, and SOD activities were measured following the standard procedure of the kits (Comin, Suzhou, China). A conductivity analyzer (INESA DDSI-308A, Shanghai, China) was used to analyze REC. Each sample weighed 0.5 g and was soaked into 10 mL of distilled water for 12 h, marked conductivity for R1, then boiled for 30 min and cooled to room temperature as R2. REC = R1/R2 × 100%.

### 4.5. RNA Isolation, cDNA Synthesis, and qRT-PCR Analysis

Total RNA was extracted using an RNAprep Pure Plant Plus Kit (Tiangen Biotech, Beijing, China), and 1 μg of the sample was reversed to cDNA with RevertAid Reverse Transcriptase (Thermo Scientific, Waltham, MA, USA) as follows: (1) Oligo(dT) ^18^ primer 1 µL, total RNA 1 ng, with water to 12 µL, 65 °C for 5 min. (2) Added the components in the indicated order, 5× Reaction Buffer 4 µL, RiboLock RNase Inhibitor (20 U/µL) 1 µL, 10 mM dNTP Mix 2 µL, RevertAid M-MuLV RT (200 U/µL), 1 µL, total volume was 20 µL, 42 °C for 60 min, then 70 °C for 5 min. qRT-PCR was conducted using CFX96 (Bio-Rad, Hercules, CA, USA) with the following steps: 8.2 µL of ddH_2_O, 0.4 µL of forward primer, 0.4 µL of reverse primer, 10.4 µL of cDNA, and 2 × 10 µL of ChamQ Universal SYBR qPCR Master Mix (Vazyme, Nanjing, China) were added into one reaction solution with real-time PCR. The following reaction steps were used: 95 °C for 30 s to denature; 5 °C for 10 s and 60 °C for 30 s, up to 40 cycles; 95 °C for 15 s, 60 °C for 60 s, and 95 °C for 15 s. *HbActin* was used as a reference gene (HQ260674.1). Three biological replicates were performed for each sample. Expression levels were calculated using 2^−ΔΔCt^. Three biological repeats and three experimental replicates were carried out for each sample. Gene expression was normalized to control for unstressed expression level, which was assigned a value of 1.

### 4.6. RNA-Seq Analysis

The rubber trees in control and treatment groups were treated with water and BR, respectively. Latex samples were subjected to RNA-seq with Biomarker technologies. Three biological replicates were carried out. RNA was extracted from each sample, and concentration and integrity of RNA sample were examined by NanoDrop (Thermo Scientific, Waltham, MA, USA), Qubit 2.0 (Invitrogen, Carlsbad, CA, USA), Agilent 2100 (Agilent, Palo Alto, CA, USA), etc. Only RNA with good quality could move on to following procedures. Qualified RNA samples were processed for library construction. In order to ensure the quality of library, Qubit 2.0 and Agilent 2100 were used to examine the concentration of cDNA and insert size. Q-PCR was processed to obtain a more accurate library concentration. A library with concentration larger than 2 nM was acceptable. The qualified library was pooled based on pre-designed target data volume and then sequenced on Illumina sequencing platform. Clean data with high quality were obtained by filtering raw data, which removed adapter sequences and reads with low quality. These clean data were further mapped to pre-defined reference genome generating mapped data. Assessment on insert size and sequencing randomness were processed on mapped data as library quality control. Basic analysis on mapped data included gene expression quantification, alternative splicing analysis, novel genes prediction, and genes structure optimization. The DEGs with fold change ≥ 1.5 and *p*-value < 0.05 were selected to analyze GO and KEGG enrichment. In order to understand the GO entries that were significantly enriched compared with the whole genomic background, the Cluster Profiler was used to conduct enrichment analysis of biological processes, molecular functions, and cell components by using hypergeometric testing methods for the differential gene sets of each group. The term obtained from the enrichment results was visualized with a histogram. The function of different genes in this group could be predicted based on the information of GO functional enrichment. Generally, the significance of the functional pathway was judged by the size of the q-value, and the smaller the q-value the more significant it was. The analysis of whether differentially expressed genes had significant differences in a certain pathway (over-presentation) comprised the pathway enrichment analysis of differentially expressed genes. For pathway significance enrichment analysis, we used pathway in the KEGG database as the unit and applied the hypergeometric test to find the pathway of significant enrichment in differentially expressed genes compared with the whole-genome background. Significant enrichment of a pathway can identify the most important biochemical and metabolic pathways and signal transduction pathways involved in genes. Cluster Profiler was used to visualize the enrichment results using bubble graphs, bar graphs, and network graphs.

### 4.7. Quantification of Phytohormones

Latex with an average weight of 25 mg was taken from the control and BR-treated samples and vortexed with 1 mL isopropanol–water (80:20, *v*/*v*, with isotopically labeled internal labels) for 30 s. The samples were then homogenized at 40 Hz for 4 min and underwent ultrasonic treatment in an ice–water bath for 5 min. This was repeated three times, and the samples were left for 1 h at −20 °C. After that, they were centrifuged at 4 °C and 14,000 r/min for 15 min, 800 μL of the supernatant was extracted, and the concentrate was rotated to dry. Then, 160 μL methyl alcohol–water (50:50, *v*/*v*) was used to redissolve the samples. Finally, the supernatant was centrifuged at 4 °C and 14,000 r/min for 15 min to 0.22 μm of filtrate for analysis. We accurately weighed the corresponding amount of the standard in the volumetric bottle for preparing the standard reserve liquid of 1000 ng/mL. We took the corresponding amount of standard reserve liquid in 10 mL volumetric bottle to prepare a mixed standard solution. The standard solution was diluted to produce a series of calibration solutions successively (containing a mixture of isotopically labeled internal labels consistent with the final concentration in the sample). A UPLC system (Waters, Milford, MA, USA) with an ACQUITY UPLC HSS T3 (100 × 2.1 mm, 1.8 μm, Waters, Milford, MA, USA) liquid chromatography column was used to separate the target compounds. The liquid chromatography consisted of aqueous solution containing 0.1% formic acid in phase A and acetonitrile containing 0.1% formic acid in phase B. The temperature of the column chamber was 40 °C, the sample tray was set to 10 °C, and the sample volume was 5 μL. The SCIEX 6500 QTRAP+ triple quadrupole mass spectrometer equipped with IonDrive Turbo V ESI ion source was used for mass spectrometry in multiple reaction monitoring (MRM) mode in this research. The ion source parameters were as follows: Curtain Gas = 35 psi, IonSpray Voltage = +5500 V, −4500 V, Temperature = 550 °C, Ion Source Gas 1 = 50 psi, and Ion Source Gas 2 = 55 psi. Prior to UHPLC-MS/MS analysis, a standard solution of the target compound was introduced into the mass spectrum. For each target compound, the transition mother–daughter ion pair with the highest signal intensity and the ion pair with the best response were selected for quantitative analysis, whereas the other ion pairs were used for qualitative analysis of the target compound. All mass spectrometry data acquisition and quantitative analysis of target compounds were performed using SCIEX Analyst Work Station Software (Version 1.7.2) and Sciex OS 2.0.1. The standard curve was regression analysis by the least square method. When the weight was set to 1/x, the calibration solution recovery rate (accuracy) and correlation coefficient (R) were the best. Lower limit of detection (LLOD) was defined as the concentration of the compound corresponding to the signal-to-noise ratio of 3, and lower limit of quantitation was defined as the concentration of the compound corresponding to the signal-to-noise ratio of 10 (US FDA guideline for bioanalytical method validation). The precision of the test was assessed by the relative standard deviation (RSD) of repeated QC (quality control) sample injection. Accuracy was assessed by the recovery of the QC sample, and the recovery was the percentage value of the measured concentration and the added concentration.

### 4.8. Statistical Analysis

Three biological repeats and three experimental replicates were carried out for each experiment. The original data were sorted using Excel 365. All data are displayed as means ± standard deviation (SD) of three replicates and were analyzed with two-way ANOVA using IBM SPSS Statistics 26 software. A significance level of 0.05 was evaluated using the Duancan method. GraphPad Prism 7.0 software was used for mapping.

## 5. Conclusions

This study revealed the effect of BR signaling network on the yield and quality of latex in rubber trees under the conditions of water deficit. In conclusion, the results showed that BR improved the photosynthetic parameters, osmotic regulators, energy syntheses, and antioxidant systems of rubber trees under water deficit conditions. Additionally, BR promoted latex yield and quality by increasing the expression of genes related to natural rubber biosynthesis and latex flow. Gene Ontology and Kyoto Encyclopedia of Genes and Genomes pathway enrichment analysis identified genes involved in phytohormones in the rubber trees. In general, BR increased latex yield and quality by slowing down physiological damage and increasing gene transcription. Moreover, current research has laid the foundation for molecular diagnosis and genetic engineering for the high yield improvement of rubber trees.

## Figures and Tables

**Figure 1 ijms-24-12857-f001:**
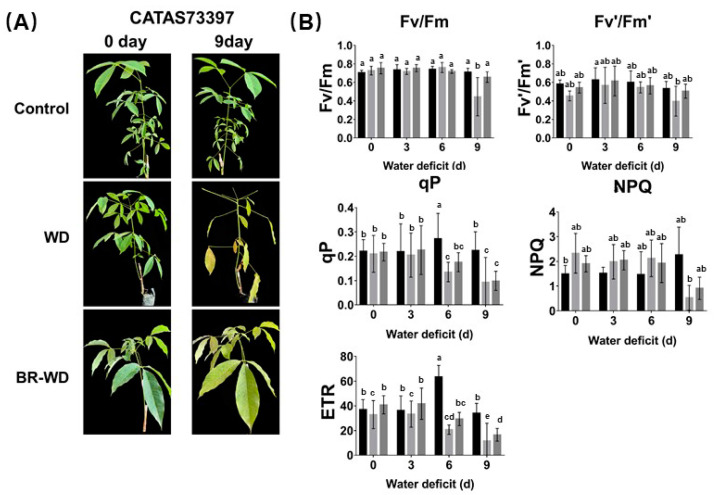
Effects of BR treatment on phenotype and photosynthesis of rubber trees under water deficit. (**A**) With water deficit 0~9 days, BR relieved leaf senescence and prevented yellowing and shedding of leaves. Control: treated with water, WD: treated with water deficit, BR-WD: treated with water deficit and BR. (**B**) Effects of exogenous BR application on photosynthesis in rubber tree leaves under water deficit stress. Different column colors represented different treatments of rubber tree, Dark: control, light grey: WD, Dark grey: BR-WD. Values represent the mean ± SD of six replicates. Bars with different letters show significant differences at the *p* < 0.05 level.

**Figure 2 ijms-24-12857-f002:**
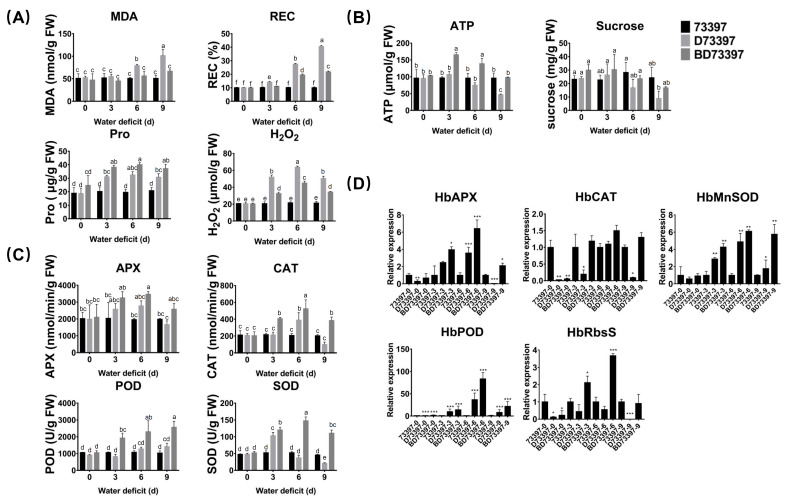
Effects of exogenous BR application on physiological index, energy biosynthesis, antioxidant system, and the gene expression of antioxidase. (**A**) The MDA, Pro, H_2_O_2_ content, and REC variation influenced by BR under a water deficit. (**B**) Energy substance content influenced by BR under a water deficit. (**C**) Antioxidant system activities in rubber trees with APX, CAT, POD, and SOD. (**D**) Effect of BR with water deficit on the expression of *HbAPX*, *HbCAT*, *HbMnSOD*, *HbPOD*, and *HbRbsS* in leaves of rubber trees. Error bars represent ± SD of three independent biological repetitions, and bars with different letters show significant differences at the *p* < 0.05 level. Bars with one star (*), two stars (**), and three stars (***) are significantly different at *p* < 0.05, *p* < 0.01, and *p* < 0.001 with *t* test, respectively.

**Figure 3 ijms-24-12857-f003:**
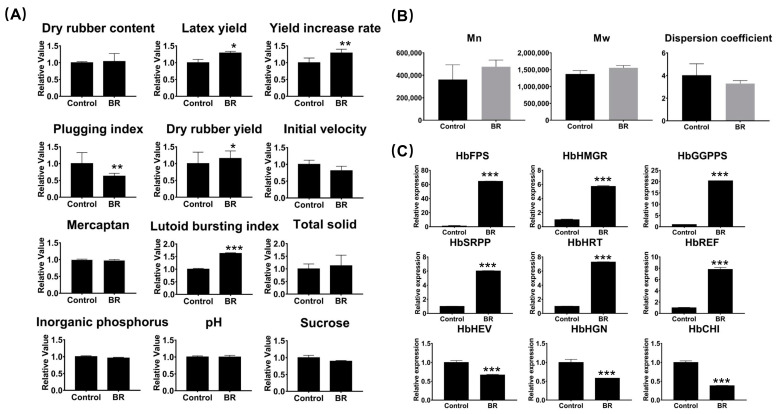
BR regulating latex yield and quality. (**A**) Variations in physiological parameters of latex after BR treatment with rubber trees. (**B**) The influence of molecular weight and dispersion coefficient related to quality of latex in rubber trees with BR treatment. (**C**) The expressions of genes related to latex yield and quality. *HbFPS*, *HbHMGR*, and *HbGGPPS* involved in the initiation of natural rubber; *HbSRPP*, *HbHRT*, and *HbREF* involved in the elongation of natural rubber; *HbHEV*, *HbHGN*, and *HbCHI* involved in latex flow. Error bars represent ± SD of three independent biological repetitions. Bars with one star (*), two stars (**), and three stars (***) are significantly different at *p* < 0.05, *p* < 0.01, and *p* < 0.001 with *t* test, respectively.

**Figure 4 ijms-24-12857-f004:**
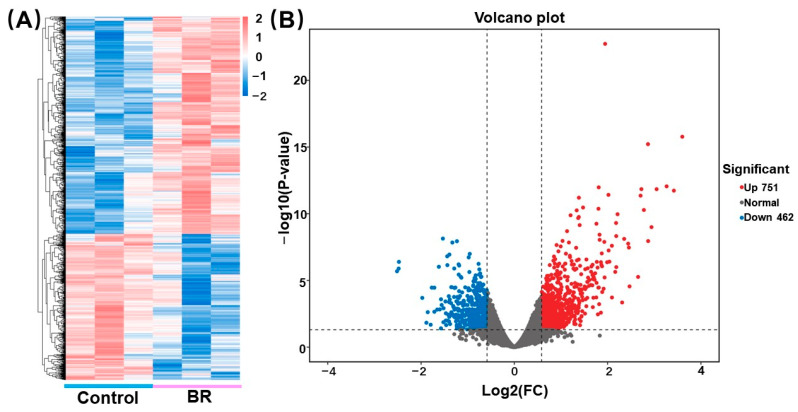
DEGs in latex with BR treatment. (**A**) Heatmaps of DEGs. The abscissa shows different treatments, the lines in blue represent control, and lines in pink represent BR treatment. The ordinate represents the genes, with red representing highly expressed genes and blue representing low-expression genes. (**B**) Volcano map of DEGs. The abscissa represents log2foldchange, and the dashed line in the transverse is a *p*-value (0.05) threshold; the ordinate represents is −log10 (*p*-value), and the two vertical dashed lines represent the two-times-expressed-difference threshold. Red dots indicate up-regulated genes, the blue bots indicate down-regulated genes, and the gray dots indicate genes with no significant differences in expression.

**Figure 5 ijms-24-12857-f005:**
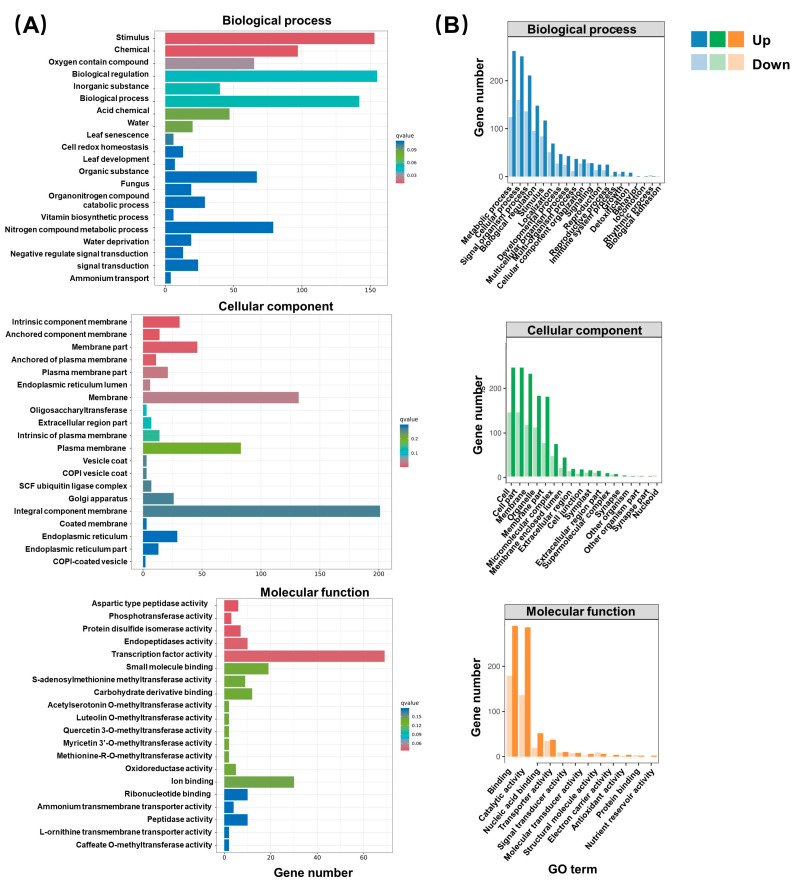
GO enrichment analysis. (**A**) GO enrichment histogram of DEGs. The abscissa represents gene number, and the ordinate represents GO terms. The color of the columns represents the q-value of the hypergeometric test. (**B**) Top GO terms. The abscissa represents the GO classification, and the ordinate represents the number of DEGs. Darker colors represent up-regulated genes, and lighter colors represent down-regulated genes.

**Figure 6 ijms-24-12857-f006:**
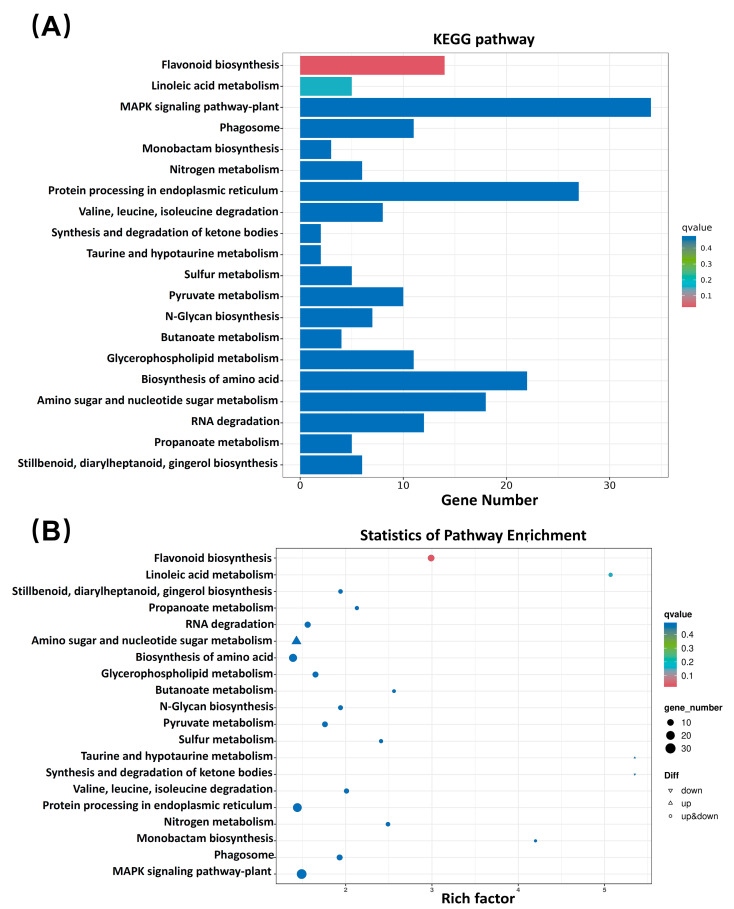
KEGG enrichment analysis. (**A**) KEGG classification histogram of DEGs. The ordinate represents the name of the KEGG pathway, and the abscissa represents the number of genes annotated to the pathway. The color of the columns represents the q-value of the hypergeometric test. (**B**) KEGG distribution map of DEGs. The abscissa represents GeneRatio, which indicates the proportion of genes of this pathway relative to all DEGs. The ordinate represents each pathway. The sizes of the dots represent the number of DEGs annotated in the pathway, and the colors of the dots represent the q-values of the hypergeometric test. Upward-pointing triangles represent up-regulated genes, and downward-pointing triangles represent down-regulated genes.

**Figure 7 ijms-24-12857-f007:**
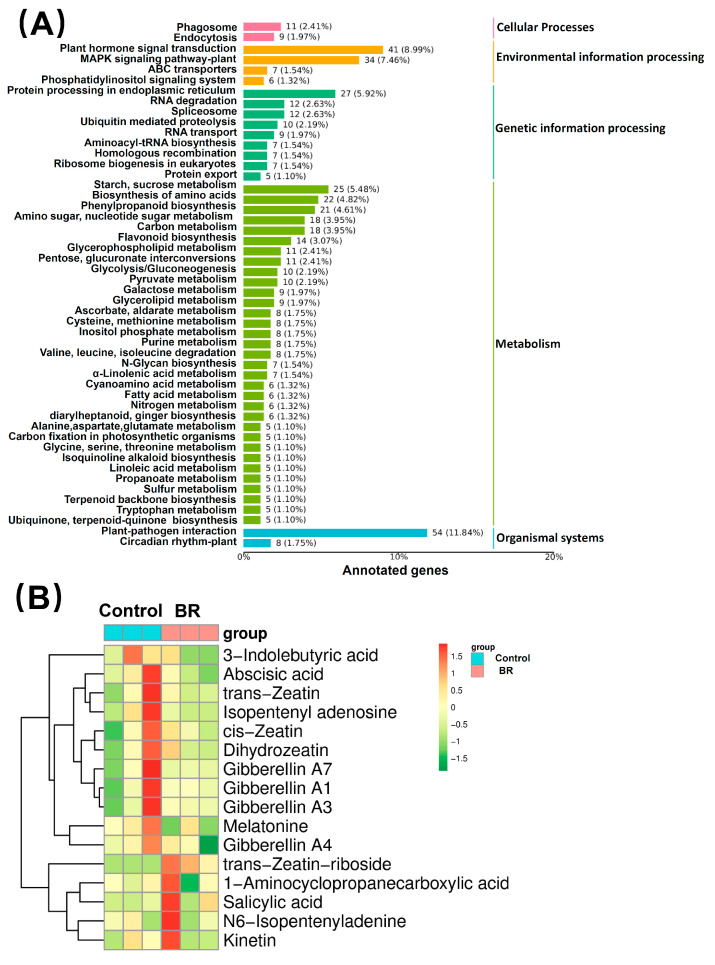
Differential metabolite enrichment. (**A**) KEGG classification map of DEGs. The ordinate represents the name of the KEGG metabolic pathway, and the abscissa represents the number of genes annotated to the pathway and their proportions. The colors of the columns represent the q-values of the hypergeometric test. (**B**) Heatmap of phytohormones. The abscissa represents each sample, the ordinate represents the quantitative value of metabolites standardized by Z-score after hierarchical clustering. The color bar above the heatmap distinguishes different groups.

## Data Availability

The origin RNA-seq data are available online at NCBI with BioProject accession number: PRJNA998206.

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
