# Peer review of "Brassinosteroids Regulate the Water Deficit and Latex Yield of Rubber Trees"

_ijms, 2023, doi:10.3390/ijms241612857_

Round 1

Reviewer 1 Report

Dear Colleagues! The presented manuscript is of great scientific and applied interest. Appropriate methods were used in the work. In the course of reading the work, there were some misunderstandings of the material. I think if you follow my comments, the work will become more understandable.

I ask you to improve the Introduction part and consider more recent articles, or at least discuss your article - reference 58.

I am attaching my suggestions.

Sincerely.

english needs to improve

Author Response

Response to the Review Comments

Dear Editors and Reviewer:

Thank you for your letter and for the reviewers’ careful reading, helpful comments, and constructive suggestions concerning our manuscript entitled ‘Brassinosteroids regulate the water deficit and latex yield of rubber trees.’ (ID: ijms-2520583), which has significantly improved the presentation of our manuscript. We have carefully considered all comments from the reviewers and revised our manuscript accordingly. The manuscript has also been double-checked. In the following section, we summarize our responses to each comment from the reviewer. We believe that our responses have well addressed all concerns from the reviewer. We hope our revised manuscript can be accepted for publication.

Reviewer 1:

Dear Colleagues! The presented manuscript is of great scientific and applied interest. Appropriate methods were used in the work. In the course of reading the work, there were some misunderstandings of the material. I think if you follow my comments, the work will become more understandable.

I ask you to improve the Introduction part and consider more recent articles, or at least discuss your article - reference 58.

I am attaching my suggestions.

Sincerely.

Response:

We gratefully appreciate for your valuable comment. Based on your comments, we revised the manuscript and responded point by point according to the file of ‘ijms-2520583-review’.

  1. line 11, we correct ‘rubber trees’ to ‘Hevea brasiliensis

  1. line 21, as the relevance to the content of the manuscript, we delete ‘plant-pathogen interaction’.

  1. line 41, ‘antioxidase’ is ‘antioxidant enzyme’, which was reference the article ‘Synthesis of Abscisic Acid in Neopyropia yezoensis and Its Regulation of Antioxidase Genes Expressions Under Hypersaline Stress.’

  1. line 57, ‘antioxidant components were studied in the work, mention them in this section’. According to your comment, we added antioxidant components illustration in this section, from line 42 to line 48 as follow: BR can alleviate oxidative damage and improve hypocotyl length, biomass, and the quality of radish sprouts stored at low temperature by improve the activity of POD and SOD. EBR improve drought tolerance of tobacco through improving SOD, POD and CAT enzyme activities and their related genes expression along with a higher accumulation of osmoregulatory substance.

  1. line 82, ‘antioxidase activity’ is another expression of ‘antioxidant enzyme activity’.

  1. line 86, ‘NR’ is the abbreviation of ‘Natural Rubber’ which is mentioned in line 68.

  1. in introduction part ‘I propose to consider in this section all the systems that are studied in the work. indicate the influence of brassinolide on the physiological and biochemical parameters of rubber. Contribute in more recent sources of literature and [58] reference’. We think this suggestion is very valuable. As brassinolide (BR) had never used in rubber trees before, so we cannot find more literatures about BR in rubber tree apart from our previous research (reference 58), so we added introduction about our work in line 75 to line 78 as follow: It is known that the receptor and transcription factors of BR signaling pathway are expressed in latex and induced by phytohormones, so we suspected that BR also has some physiological functions in rubber tree.

  1. line 167, based on your comment, we corrected the whole manuscript to change the typeface of genes in italics.

  1. in ‘Discussion’ section, ‘what methods of treatment of rubber plants with brassinolide are most effective. leaf spraying, basal treatment. and why the method you are using is chosen, it can be widely used’. Different experiments need different treatments, when hormones were used in budings and seedings, we sprayed the leaves in general. When hormones were used in breading trees, the bark is cut along the cutting line and then the treatment solution is evenly applied to the cutting surface with a brush. In this article, breading trees were used and we appended detailed description in ‘Materials and Methods’ section in line 382 to line 387.

  1. line 285 to line 295. ‘I propose to consider more clearly the mechanisms of the effect of brassinosteroido on the yield and quality of rubber. not quite clear.’. We appreciate with your kindly suggestion and we stated the mechanism effect of BR much clearer as follow: We suspect that BR improves latex yield by increasing the lutioid bursting index and reducing the plugging index by decreasing the expression level of HbHGN, HbCHI and HbHEV which prevent latex flow progress. The molecular weight of natural rubber is heterogeneous or polydisperse. The average molecular weight is used as the characterization of natural rubber molecular weight, and the molecular weight distribution is used to characterize the degree of dispersion in natural rubber dispersion, which are important parameters used to measure latex quality. BR improved the Mn but caused no difference in Mw, reducing the dispersion coefficient (represented by Mw/Mn), thus improving the quality of natural rubber. HbFPS, HbHMGR and HbGGPPS are related to NR bio-synthesis, and HbSRPP, HbHRT, HbREF are related to rubber hydrocarbon chain extension, and enhance molecular weight, which was increased by BR in latex. Physiological indicators and the expression of related genes reconfirmed our conclusions that BR improvs the yield and quality of latex. From line 329 to line 342.

  1. line 337 ‘to improve understanding, add how you treated plants with hormone’. Thank you for your valuable opinion and we expanded on the operation steps in ‘Materials and Methods’ section in line 382 to line 387.

To sum up, we have made corrected modifications on the revised manuscript. Please do not hesitate to contact us if there are any question. Thanks again to the reviewer and editor for your hard work! Best wishes to you!

Authors: Bingbing Guo, Mingyang Liu, Hong Yang, Longjun Dai, Lifeng Wang

Reviewer 2 Report

The study by Guo et al. is an attempt to characterise the effects of brassinosteroid treatment on various aspects of latex production and physiology in rubber trees.  The text is very difficult to read due to numerous language problems, including grammatical errors (such as poorly structured sentences), typographical errors and awkward word choice.

Materials and methods are also poorly described, and essential details for understanding and reproducing the bioinformatics analyses are not disclosed. As an example, the article does not provide information on the software used to align and process the reads or to quantify expression levels. No information is provided on the programs used to detect differentially expressed genes or to study the enrichment of gene ontology terms or KEGG pathways.

The authors have also not deposited their raw data from the RNA-Seq experiments in publicly available databases, making it impossible to assess their quality or to reproduce the analyses presented. The data availability statement is incomplete and unacceptable in its current form.

For all of the above reasons, and particularly because of the serious problems in reading and interpreting the text in its current form, I cannot recommend it for publication in the journal.

Please see above

Author Response

Response to the Review Comments

Dear Editors and Reviewer:

Thank you for your letter and for the reviewers’ careful reading, helpful comments, and constructive suggestions concerning our manuscript entitled ‘Brassinosteroids regulate the water deficit and latex yield of rubber trees.’ (ID: ijms-2520583), which has significantly improved the presentation of our manuscript. We have carefully considered all comments from the reviewers and revised our manuscript accordingly. The manuscript has also been double-checked. In the following section, we summarize our responses to each comment from the reviewer. We believe that our responses have well addressed all concerns from the reviewer. We hope our revised manuscript can be accepted for publication.

Reviewer 2:

The study by Guo et al. is an attempt to characterise the effects of brassinosteroid treatment on various aspects of latex production and physiology in rubber trees. The text is very difficult to read due to numerous language problems, including grammatical errors (such as poorly structured sentences), typographical errors and awkward word choice.

Materials and methods are also poorly described, and essential details for understanding and reproducing the bioinformatics analyses are not disclosed. As an example, the article does not provide information on the software used to align and process the reads or to quantify expression levels. No information is provided on the programs used to detect differentially expressed genes or to study the enrichment of gene ontology terms or KEGG pathways.

The authors have also not deposited their raw data from the RNA-Seq experiments in publicly available databases, making it impossible to assess their quality or to reproduce the analyses presented. The data availability statement is incomplete and unacceptable in its current form.

For all of the above reasons, and particularly because of the serious problems in reading and interpreting the text in its current form, I cannot recommend it for publication in the journal.

Response:

We gratefully appreciate for your valuable comments. Based on your comments, we revised the manuscript and responded point by point. The revision instructions are as follows:

  1. The text is very difficult to read due to numerous language problems, including grammatical errors (such as poorly structured sentences), typographical errors and awkward word choice.

Reply: Thank you for taking time out of your busy schedule to review the manuscript, we have made meticulous modifications to this manuscript, and polished by MAPI English editing service, therefore, the readers may understand our work more clearly. The corrected details are listed as tracked in the manuscript.

  1. Materials and methods are also poorly described, and essential details for understanding and reproducing the bioinformatics analyses are not disclosed. As an example, the article does not provide information on the software used to align and process the reads or to quantify expression levels. No information is provided on the programs used to detect differentially expressed genes or to study the enrichment of gene ontology terms or KEGG pathways.

Reply: We are very grateful to your comments for the manuscript. According to your advice, we refined the ‘Materials and methods’ section in manuscript and presented in the revised manuscript.

  1. The authors have also not deposited their raw data from the RNA-Seq experiments in publicly available databases, making it impossible to assess their quality or to reproduce the analyses presented. The data availability statement is incomplete and unacceptable in its current form.

Reply: It is very kind of you to point out our cursoriness. In fact, when received your comments, we upload our raw data of RNA-seq to NCBI database immediately and appended the accession number in ‘Data Availability Statement’ section in line 516 to line 518 as follows: ‘The origin RNA-seq data are available online at NCBI with BioProject accession number: PRJNA998206.’

To sum up, we have made corrected modifications on the revised manuscript. Please do not hesitate to contact us if there are any question. Thanks again to the reviewer and editor for your hard work! Best wishes to you!

Authors: Bingbing Guo, Mingyang Liu, Hong Yang, Longjun Dai, Lifeng Wang

Reviewer 3 Report

The text below contains comments on manuscript entitled “Brassinosteroids regulate the water deficit and latex yield of rubber trees”

The manuscript is focused on treatment of rubber tree with hormones, such as brassinosteroids and improve the photosynthetic index parameter, antioxidase avtivity, osmotic regulator to alleviate the water deficit of rubber trees. Along with that the treatment increased the expression of natural rubber synthetic gene and decreased the expression of latex plugging gene.

The manuscript is well written, with logically structured experimental design, the results of which fully explain the aim of the study. I would just advice the authors to go once again through the manuscript for minor English corrections and think about to present in bigger size the charts on figures 1, 2 and 3.

 Minor editing of English language required

Author Response

Response to the Review Comments

Dear Editors and Reviewer:

Thank you for your letter and for the reviewers’ careful reading, helpful comments, and constructive suggestions concerning our manuscript entitled ‘Brassinosteroids regulate the water deficit and latex yield of rubber trees.’ (ID: ijms-2520583), which has significantly improved the presentation of our manuscript. We have carefully considered all comments from the reviewers and revised our manuscript accordingly. The manuscript has also been double-checked. In the following section, we summarize our responses to each comment from the reviewer. We believe that our responses have well addressed all concerns from the reviewer. We hope our revised manuscript can be accepted for publication.

Reviewer 3:

The text below contains comments on manuscript entitled “Brassinosteroids regulate the water deficit and latex yield of rubber trees”

The manuscript is focused on treatment of rubber tree with hormones, such as brassinosteroids and improve the photosynthetic index parameter, antioxidase avtivity, osmotic regulator to alleviate the water deficit of rubber trees. Along with that the treatment increased the expression of natural rubber synthetic gene and decreased the expression of latex plugging gene.

The manuscript is well written, with logically structured experimental design, the results of which fully explain the aim of the study. I would just advice the authors to go once again through the manuscript for minor English corrections and think about to present in bigger size the charts on figures 1, 2 and 3.

Response:

Thank you for taking time out of your busy schedule to review the manuscript, we have made meticulous modifications to this manuscript, and polished by MAPI English editing service, therefore, the readers may understand our work more clearly. The corrected details are listed as heighted in the manuscript. What’s more, we had larger all the graphs and made the letters with bold for more clearer in the revised manuscript.

To sum up, we have made corrected modifications on the revised manuscript. Please do not hesitate to contact us if there are any question. Thanks again to the reviewer and editor for your hard work! Best wishes to you!

Authors: Bingbing Guo, Mingyang Liu, Hong Yang, Longjun Dai, Lifeng Wang

Reviewer 4 Report

The use of brassinosteroids as an anti-stress growth regulator is still relevant, as not all mutual interactions with other growth regulators and increases/decreases under the influence of stressors are yet known. The manuscript meets the conditions for publication in the specified journal. The manuscript is relatively well written, but some parts need to be edited. In the methodology, it is not entirely clear how the plants were grown. In other words, on which medium (soil, sand, hydroponics) were the plants grown? The text mentions net photosynthesis as one of the parameters, but only chlorophyll fluorescence is mentioned in the methodology. Please provide an explanation or addition. The graphs presented in the text are small and therefore not very clear. Please review them. Are there correlations between the monitored parameters? It is necessary to unify the citations. Are all older sources relevant?  

Author Response

Response to the Review Comments

Dear Editors and Reviewer:

Thank you for your letter and for the reviewers’ careful reading, helpful comments, and constructive suggestions concerning our manuscript entitled ‘Brassinosteroids regulate the water deficit and latex yield of rubber trees.’ (ID: ijms-2520583), which has significantly improved the presentation of our manuscript. We have carefully considered all comments from the reviewers and revised our manuscript accordingly. The manuscript has also been double-checked. In the following section, we summarize our responses to each comment from the reviewer. We believe that our responses have well addressed all concerns from the reviewer. We hope our revised manuscript can be accepted for publication.

Reviewer 4:

The use of brassinosteroids as an anti-stress growth regulator is still relevant, as not all mutual interactions with other growth regulators and increases/decreases under the influence of stressors are yet known. The manuscript meets the conditions for publication in the specified journal. The manuscript is relatively well written, but some parts need to be edited. In the methodology, it is not entirely clear how the plants were grown. In other words, on which medium (soil, sand, hydroponics) were the plants grown? The text mentions net photosynthesis as one of the parameters, but only chlorophyll fluorescence is mentioned in the methodology. Please provide an explanation or addition. The graphs presented in the text are small and therefore not very clear. Please review them. Are there correlations between the monitored parameters? It is necessary to unify the citations. Are all older sources relevant?

Response:

We gratefully appreciate for your valuable comment. Based on your comments, we revised the manuscript and responded point by point.

  1. In the methodology, it is not entirely clear how the plants were grown. In other words, on which medium (soil, sand, hydroponics) were the plants grown?

Reply: thank you for your kindly remind and we have revised the description in line 373 ‘Rubber tree variety ‘CATAS73397’ buddings were grown in the plastic pots with soil at experimental farm in the Chinese Academy of Tropical Agricultural Sciences in Danzhou city, Hainan province, China’

  1. The text mentions net photosynthesis as one of the parameters, but only chlorophyll fluorescence is mentioned in the methodology. Please provide an explanation or addition.

Reply: Thank you point out our negligence. We had added the description in line 420 to line 428 as follows: Maximum fluorescence and initial minimum fluorescence were measured in dark-adapted leaves to calculate the maximum photochemical efficiency (Fv/Fm) of photosystem II. The numerical value of Fo, Fm (maximal fluorescence yield), Fo’ (initial fluorescence under light adaptation), Fm’ (maximum fluorescence under light adaptation), Fv/Fm (PS II), ETR (relative electron transport rate), Y(II) (actual quantum yield of PS II), Y(NPQ) (regulatory dissipation) and qL (photochemical quenching) were measured with PAM-2500 high-performance chlorophyll fluorometer. Other parameters were calculated based on the measured values.

  1. The graphs presented in the text are small and therefore not very clear. Please review them.

Reply: Thank you for your good suggestion to improve our manuscript. We had larger the graphs and made the letters with bold for more clearer in the revised manuscript.

  1. Are there correlations between the monitored parameters? It is necessary to unify the citations. Are all older sources relevant?

Reply: We are grateful for the suggestion. We supplemented the correlation between the monitored parameters in the supplement materials. And we unified the citations with replace the older sources to latest relevant sources.

To sum up, we have made corrected modifications on the revised manuscript. Please do not hesitate to contact us if there are any question. Thanks again to the reviewer and editor for your hard work! Best wishes to you!

Authors: Bingbing Guo, Mingyang Liu, Hong Yang, Longjun Dai, Lifeng Wang

Round 2

Reviewer 1 Report

Accepted in present form

minor improvement in English

Author Response

Dear Editors and Reviewer:

Thank you for your letter and for the reviewers’ careful reading, helpful comments, and constructive suggestions concerning our manuscript entitled ‘Brassinosteroids regulate the water deficit and latex yield of rubber trees.’ (ID: ijms-2520583), which has significantly improved the presentation of our manuscript. We have carefully considered all comments from the reviewers and revised our manuscript accordingly. The manuscript has also been double-checked and polished by MDPI english editing service. We believe that our responses have well addressed all concerns from the reviewer. We hope our revised manuscript can be accepted for publication.

Sincerely.

Bingbing Guo, Mingyang Liu, Hong Yang, Longjun Dai, Lifeng Wang
